# Assessing the Use of BotulinumtoxinA for Hyperactive Urinary Tract Dysfunction a Decade after Approval: A Single-Blind Study to Evaluate the Reduction in Pain in OnabotulinumtoxinA Detrusor Injection Using Different Injection Needles

**DOI:** 10.3390/toxins16090395

**Published:** 2024-09-14

**Authors:** Heinrich Schulte-Baukloh, Catarina Weiss, Thorsten Schlomm, Sarah Weinberger, Hendrik Borgmann, Dirk Höppner, Kathrin Haberecht, Jörg Neymeyer

**Affiliations:** 1Department of Urology, Charité—University Hospital Berlin, 10117 Berlin, Germany; thorsten.schlomm@charite.de (T.S.); sarah.weinberger@charite.de (S.W.); joerg.neymeyer@charite.de (J.N.); 2Urologic Practice Turmstrasse, 10551 Berlin, Germany; dh@urologie-turmstrasse.de (D.H.); kh@urologie-turmstrasse.de (K.H.); 3Department of Urology, University Hospital Brandenburg, 14770 Brandenburg, Germany; hendrik.borgmann@uk-brandenburg.de; 4Urologic Practice, Kurfürstendamm 139, 10711 Berlin, Germany; cweiss@urologie-weiss.de

**Keywords:** overactive bladder, botulinumtoxinA detrusor injection, needle thickness, pain, local anesthesia

## Abstract

Overactive bladder (OAB) has a significant impact on the quality of life; thus, it requires treatment that can be adhered to over a long period without undue side effects. The current treatment which uses an anticholinergic or β-3 agonist may fail to improve symptoms and has side effects, leading to high discontinuation rates. OnabotulinumtoxinA (OnabotA) detrusor injection has been approved for idiopathic OAB as a second-line treatment with good effectiveness and tolerability. This study used a visual analog scale (VAS) to assess the impact of the type of needle used for OnabotA detrusor injections under local anesthesia on the pain levels after each injection. This study included 68 female patients. We used three different needles with thicknesses ranging from 22 to 27 gauge, lengths between 4 and 5 mm, and different cuts. The sensation of pain was rated at each standardized injection location. Regardless of the needle used, the patients’ perceptions of pain at the beginning of the procedure were rated as being less than the subsequent injections. Most pain sensations were rated as low to moderate. The mean pain sensation on the VAS was 2.5 ± 0.3 overall, i.e., for all patients and needles used. Statistically significant differences in pain sensation were rated only at some locations of the bladder (on the back wall and the right side of the bladder). The single needles averaged the following pain scores: 2.8 ± 0.3 for needle A (20 G, 4 mm), 2.1 ± 0.3 for needle B (27 G, 5 mm), and 2.6 ± 0.4 for needle C (20 G, 4 mm, sharp cut 15°). The 27-gauge needle caused significantly less pain, and it had no negative impact due to its length, which was 1 mm longer than the other needles. Thus, the needle thickness was a decisive factor in the patients’ perceptions of pain.

## 1. Introduction

Overactive bladder (OAB) is characterized by symptoms of urinary urgency, with or without urgency urinary incontinence, usually with frequency and/or nocturia, in the absence of a proven infection or other pathology [1]. The symptoms can be considerable and can significantly limit the quality of life for both sexes in many areas of life [2,3]. Additionally, the economic costs are enormous [4].

Accordingly, there is a wide range of approaches to treat OAB syndrome therapeutically. Conservative therapeutic measures include behavioral therapeutic approaches, such as weight reduction, caffeine reduction, smoking cessation, or fluid management, and pelvic floor therapy, such as pelvic floor training and electrical stimulation including tibial nerve stimulation [5].

Drug therapy is a primary approach to the treatment of OAB [6]. First-line pharmacological approaches to OAB consist of anticholinergic drugs, such as oxybutynin, trospium chloride, tolterodine, solifenacin, darifenacin, and fesoterodine (in chronological order). These act by competitively inhibiting the activation of muscarinic receptors of the detrusor muscle by acetylcholine [7]. Although usually effective, these drugs have many side effects (e.g., dry throat, blurred vision, constipation, and other gastrointestinal complaints), which, in addition to possible ineffectiveness, lead to a significant proportion of therapy discontinuations. They have been demonstrated to cause cognitive limitations and the development of dementia in older patients [8]. The discontinuation rate of these medications increases to 80% after one year [9].

Alternatively, mirabegron or vibegron, strong and selective β3-adrenoceptor agonists, have been available for several years and have good efficacy, a lower rate of side effects, and correspondingly better adherence to therapy [9]. OnabotulinumtoxinA detrusor injection (OnabotA-DI) has been approved for idiopathic OAB (iOAB) for a decade as a second-line treatment after unsuccessful anticholinergic or β3-mimergic therapy and is recommended in the EAU guidelines [5], becoming firmly established in patient settings. OnabotA effectively blocks the release of various neurotransmitters, such as acetylcholine, ATP, or substance P, and leads to a reduction in certain ion channels and pain receptors (TRPV1 and P2X3), which explains the reduction in contractility, urinary bladder desensitization, a reduction in urgency, and also a reduction in pain [10]. A number of studies have proven the effectiveness of OnabotA detrusor injections (DIs). Initial studies found the injection of 200 U OnabotA to be effective [11], but Dmochowski et al. conducted a first optimized dose-finding study, which led to the dosage of 100 U OnabotA that is used to this day [12].

The approval study with 100 U of OnabotA demonstrated good effectiveness in alleviating OAB symptoms and urinary incontinence, in quality of life, and in urodynamic parameters with good tolerability [13,14,15].

Since then, the medication has been applied to the lateral and posterior walls of the bladder at 20 different points, each with 0.5 mL of OnabotA reconstituted with NaCl (equivalent to 5 U per injection site), excluding the trigone. Most users, however, include the trigone as an injection site, and scientific data support this approach. For example, a review by Jo et al. [16] summarizes comparisons of trigone-including versus trigone-sparing injections, which indicate that including the trigone reduces the detrusor pressure and increases the volume at first desire to void. The depth of injection, intradetrusor or suburothelial, does not influence the efficacy or safety of OnabotA, as described by Jo et al. and in other studies [16,17].

For many experts, a challenge in outpatient use seems to be the fear of pain and an insufficient effect of local anesthesia [18]. Several working groups have addressed the problem of pain reduction, and protocols such as adding sodium bicarbonate to local anesthesia, reducing the number of injection sites, or exposure to EMDA (electromotive drug administration) therapy have resulted in better acceptance of the therapy [19]. The use of a rigid instrument versus a flexible instrument may also influence pain, although to our knowledge, there are no study data in this regard.

The important topic of patients’ acceptance of OnabotA injections through reduced pain has been addressed not only in urology, but also in several therapeutic areas, e.g., by Nasser et. al. [20] in dermatology for the treatment of palmar and plantar hyperhidrosis. For this indication, topical anesthesia, ice, and vibration are the safest and most convenient noninvasive methods to relieve pain associated with botulinum detrusor injection. Aesthetic medicine faces similar problems: Athadeu et al. compared the effectiveness of topical anesthesia cream, vibratory stimulus, cryotherapy, pressure, and no intervention for reducing pain during and immediately after injections into the forehead [21]. There was no real “break-through” favorite in esthetics and dermatology to make the therapy more pleasant; no analgesic method to reduce pain was superior to any others. In contrast, Sezgin et al. [22], again in esthetic medicine, were able to achieve at least partial pain relief by using different needle gauges; in their study, an assessment of the multiple-injection process demonstrated a significant difference in pain level, favoring their very thin 33 G needle.

Motivated by such studies to transfer these insights to our urology field, our study aims to evaluate whether diverse levels of pain occur when using different injection needles. Therefore, our main hypothesis is that in the context of OnabotA-DI, there is a different degree of pain when using different needles available on the international urology market with different needle thicknesses, needle lengths, and needle cut shapes. The degree of pain should be determined using a VAS pain score. The needles examined in our study are listed anonymously because the user’s acceptance does not depend exclusively on the pain experienced by the patient, but also on practicality and handling in use; the length, elasticity, and maneuverability of the cannula; and the cost of the needles, to name a few.

## 2. Results

This study included 68 female patients. The patients’ characteristics in the individual randomized needle groups showed no statistical differences with regard to age, diagnosis, and frequency of injections (Table 1). Eighty-eight percent of patients suffered from iOAB. In eight patients, the injection was carried out despite a positive urine culture because the pronounced urge incontinence was the cause of the recurrent urinary tract infections. However, in these patients, there were no relevant symptoms (fever/pain) that would otherwise have been exclusion criteria.

Regardless of the needle used, there was a trend in the patients’ perceptions of pain, where the pain at the beginning of the procedure (first 1–3 injections) was classified as less intense than the subsequent injections, which had a sense of an increasing sensation of pain or pain aggravation. Otherwise, the sensation of pain could generally be classified as low to moderate regardless of the needle used; the mean pain sensation on the VAS (scale of 1–10) was 2.5 ± 0.3, and the single needles averaged the following pain scores: 2.8 ± 0.3 for needle A, 2.1 ± 0.3 for needle B, and 2.6 ± 0.4 for needle C. Although there appeared to be consistent differences in pain sensation between needles (see Table 2), statistically significant differences were found only in some areas of the urinary bladder (see Table 2 and Table 3). The needle that caused significantly less pain was the 27-gauge needle, whose 1 mm longer length compared to the other needles had no negative impact. Needle thickness was the decisive factor in the patients’ perceptions of pain. The different pointed 15° cut of the 22-gauge needle (needle C) also contributed to pain reduction (see comparison to the 22-gauge needle (needle A) in Table 3). Needle B (27-gauge needle) showed a very mild advantage in comparison to needle C (22 gauge, pointed 15° cut).

**Table 2 toxins-16-00395-t002:** Pain scales (VAS 1–10) at the individual locations of the urinary bladder.

Characteristic	Overall n = 68	Needle A (22 G, 4 mm) n = 33 ^2^	Needle B (27 G, 5 mm) n = 20 ^2^	Needle C (22 G, 4 mm, Sharp Cut 15°) n = 15 ^2^	*p*-Value ^1^
** LS1 **					0.3
Mean (SD)	1.7 (1.1)	1.9 (1.2)	1.5 (0.7)	1.6 (1.1)	
Range	1.0–5.0	1.0–5.0	1.0–3.0	1.0–5.0	
** LS2 **					0.15
Mean (SD)	2.2 (1.5)	2.5 (1.6)	1.7 (0.8)	2.4 (1.6)	
Range	1.0–7.0	1.0–7.0	1.0–4.0	1.0–7.0	
** LM1 **					0.15
Mean (SD)	2.6 (1.8)	2.8 (2.1)	1.9 (1.1)	2.8 (1.7)	
Range	1.0–9.0	1.0–9.0	1.0–5.0	1.0–6.0	
** LM2 **					0.14
Mean (SD)	2.9 (1.7)	3.0 (1.9)	2.4 (1.3)	3.5 (1.6)	
Range	1.0–9.0	1.0–9.0	1.0–5.0	1.0–6.0	
** LM3 **					0.5
Mean (SD)	2.7 (1.7)	2.9 (1.9)	2.4 (1.3)	2.5 (1.6)	
Range	1.0–10.0	1.0–10.0	1.0–4.0	1.0–7.0	
** M1 **					0.030
Mean (SD)	2.6 (1.9)	3.3 (2.3)	2.2 (1.3)	1.9 (1.2)	
Range	1.0–10.0	1.0–10.0	1.0–5.0	1.0–5.0	
** M2 **					0.2
Mean (SD)	2.7 (1.7)	2.9 (2.0)	2.2 (1.1)	3.0 (1.5)	
Range	1.0–8.0	1.0–8.0	1.0–5.0	1.0–6.0	
** M3 **					0.4
Mean (SD)	2.5 (1.7)	2.5 (1.6)	2.2 (1.4)	3.0 (2.4)	
Range	1.0–8.0	1.0–6.0	1.0–6.0	1.0–8.0	
** Top **					0.4
Mean (SD)	2.5 (1.5)	2.7 (1.5)	2.2 (1.2)	2.6 (1.6)	
Range	1.0–7.0	1.0–6.0	1.0–5.0	1.0–7.0	
** RM1 **					0.6
Mean (SD)	2.6 (1.7)	2.7 (1.7)	2.3 (1.3)	2.7 (2.1)	
Range	1.0–7.0	1.0–7.0	1.0–6.0	1.0–7.0	
** RM2 **					0.7
Mean (SD)	2.5 (1.6)	2.7 (1.6)	2.3 (1.6)	2.5 (1.7)	
Range	1.0–7.0	1.0–7.0	1.0–7.0	1.0–6.0	
** RM3 **					0.2
Mean (SD)	2.5 (1.6)	2.8 (1.8)	1.9 (1.4)	2.5 (1.3)	
Range	1.0–8.0	1.0–8.0	1.0–6.0	1.0–6.0	
** RS1 **					0.057
Mean (SD)	2.4 (1.7)	2.7 (1.7)	1.7 (1.1)	2.9 (2.2)	
Range	1.0–9.0	1.0–7.0	1.0–4.0	1.0–9.0	
** RS2 **					0.3
Mean (SD)	2.4 (1.6)	2.7 (1.7)	2.0 (1.6)	2.4 (1.3)	
Range	1.0–8.0	1.0–7.0	1.0–8.0	1.0–5.0	
** RBB **					0.7
Mean (SD)	2.2 (1.7)	2.4 (1.8)	2.0 (1.4)	2.3 (1.8)	
Range	1.0–8.0	1.0–8.0	1.0–7.0	1.0–6.0	
** LBB **					0.4
Mean (SD)	2.6 (2.0)	3.0 (2.3)	2.2 (1.5)	2.4 (1.7)	
Range	1.0–10.0	1.0–10.0	1.0–7.0	1.0–6.0	
** Trigone **					0.12
Mean (SD)	2.8 (2.1)	3.3 (2.5)	2.1 (1.2)	3.0 (2.2)	
Range	1.0–9.0	1.0–9.0	1.0–6.0	1.0–9.0	

The injection locations and the associated pain sensations (mean, standard deviation, and range) on the VAS (1 to 10) (for the injection scheme, please see Figure 1): left side (LS) 1 + 2, left-middle (LM) 1 + 2 + 3, middle (M) 1 + 2 + 3, top right-middle (RM) 1 + 2 + 3, right side (RS) 1 + 2, right bladder base (RBB), left bladder base (LBB), trigone (TRI). ^1^ Based on one-way ANOVA. ^2^ There were different frequencies of needles due to delivery bottlenecks during and after the COVID-19 pandemic (see text).

**Table 3 toxins-16-00395-t003:** Pairwise comparisons of *significant* scales.

Characteristic	Needle A (22 G, 4 mm) N = 33 ^1^	Needle B (27 G, 5 mm) N = 20 ^1^	Needle C (22 G, 4 mm, Sharp Cut 15°) N = 15 ^1^	Needle A vs. Needle B	Needle A vs. Needle C	Needle B vs. Needle C
** M1 **				** 0.045 **	** 0.019 **	0.6
Mean (SD)	3.29 (2.33)	2.20 (1.28)	1.86 (1.23)			
** RM3 **				** 0.050 **	0.6	0.3
Mean (SD)	2.80 (1.83)	1.90 (1.45)	2.50 (1.29)			
** RS1 **				** 0.040 **	0.7	** 0.036 **
Mean (SD)	2.70 (1.68)	1.70 (1.08)	2.93 (2.16)			

The injection locations where the differences in pain sensation reached statistical significance, marked in **bold** letters (middle M1, right-middle RM3, right side RS 1); for injection scheme, please see Figure 1. ^1^ There were different frequencies of needles due to delivery bottlenecks during and after the COVID-19 pandemic (see text).

**Figure 1 toxins-16-00395-f001:**
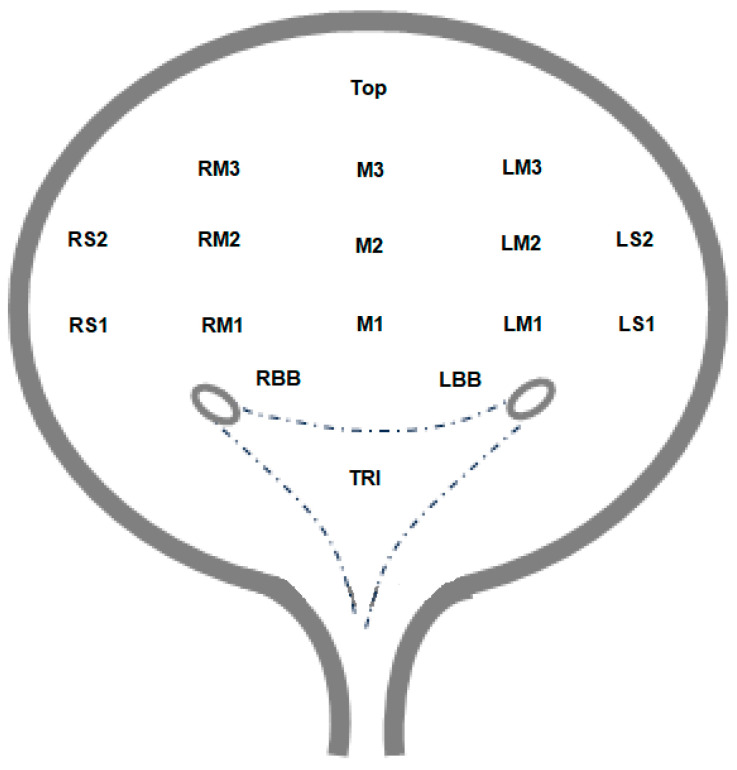
Injection scheme, which was reduced to 17 injections: left side (LS) 1 + 2, left-middle (LM) 1 + 2 + 3, middle (M) 1 + 2 + 3, top right-middle (RM) 1 + 2 + 3, right side (RS) 1 + 2, right bladder base (RBB), left bladder base (LBB), trigone (TRI).

## 3. Discussion

Many studies have shown the effectiveness of OnabotA detrusor injection for the treatment of neurogenic and non-neurogenic urinary bladder dysfunction with an acceptable pattern of side effects [23,24]. One drawback, however, is the level of pain caused by injection therapy, which varies greatly from person to person.

In our study protocol, 17 injections were carried out instead of the 20 injections recommended in the package insert, with the knowledge that a smaller number of injections might lead to an overall lower level of pain. As discussed in the following paragraph, the potential pain-relieving influence of sodium bicarbonate when added to the lidocaine solution was *not* yet considered in our protocol (although we followed this recommendation after this study). The injection pattern included the injection of the trigone. Among these injection parameters mentioned, the mean pain sensation in our study was 2.5 on the VAS of 1–10, and there were significant differences between the needles used at some injection sites. At first glance, the pain intensities in Table 2 appear (for example, between needles A and B) fairly constant in favor of needle B. However, the differences were only statistically significant at a few injection sites, which we can best explain by the underpowered study due to the supply restraints of some needles during and after the COVID-19 pandemic. However, the differences appear to be of moderate clinical importance because the inter-needle range between the average values was relatively small, namely VAS 2.1 to 2.8. In contrast, the range of individual pain perception is substantial. Pain levels between 1 and the maximum of 10 were reported, although high pain levels were reported only selectively. This is probably related to hitting a highly innervated pain point. Nevertheless, for patients who repeatedly rated the pain as being severe and stressful, the implementation of this therapy under local anesthesia should be carefully reconsidered and, if necessary, general anesthesia should be offered for the procedure.

An important question arises from our experiences and those of other users: how can we reduce the pain of OnabotA detrusor injection and thus increase the acceptance of this therapy? The literature discusses different approaches to this.

In neurology, Dressler et al. were able to show in the treatment of blepharospasm that the pain level depends significantly on the pH values of the injection fluid: by using Ringer’s acetate solution instead of normal saline when reconstituting OnabotA, the sensation of pain could be significantly reduced [25].

Accordingly, similar approaches to influencing the pH of the anesthetic solution in particular have been chosen in urology, and quite different results were obtained: Pereira e Silva et al. performed a double-blind, randomized controlled trial comparing pre-injection intravesical instillations of 20 mL 2% lidocaine + 10 mL 8.4% sodium bicarbonate with 20 mL 2% lidocaine + 10 mL 0.9% saline solution. The study was carried out in women (86.2%) and men, with idiopathic detrusor overactivity (73.3%) and with neurogenic detrusor overactivity (18.1%) and, to a small extent, in patients with bladder pain syndrome. Subjects who received alkalinized lidocaine (AL) solution reported lower pain scores immediately after the procedure than those who received lidocaine solution with saline (numeric rating scale [NRS], 2.37 ± 0.31 and 4.44 ± 0.36, respectively; *p* < 0.01) [26].

Other authors were unable to demonstrate this beneficial effect of alkalinization. A randomized comparison between AL solution (10 mL 8.4% sodium bicarbonate + 20 mL 2% lidocaine solution + 22 mL sterile Aquagel) and lidocaine gel (LG) (22 mL standard 2% lidocaine gel + 30 mL 0.9% normal saline solution) showed that using AL solution for anesthesia is not superior to lidocaine gel during intra-vesical OnabotA injections [27].

Kocher et al. compared different lidocaine instillations including AL in a prospective study involving 25 patients regarding the effect on pain. There was no statistically significant change in patient-reported discomfort (using the visual analog scale [VAS]) for different lidocaine instillations (*p* = 0.913) nor for the instillation dwell time (*p* = 0.14) [28].

Another study by Steward et al. showed that oral analgesia alone with 200 mg of oral phenazopyridine taken 1–2 h before the procedure was non-inferior for procedural pain control compared to intravesical instillation with 50 mL of 2% lidocaine instilled 20 min before the procedure [29].

An interesting approach to reduce pain through intravesical OnabotA treatment was described by Ladi-Seyedian et al. in children with neurogenic urinary bladder. A comparison was made between the application of 10 U/kg body weight of AbobotA (Dysport/Ipsen) via either injection therapy at 40 sites of the detrusor muscle and leaving the drug in the same dosage in the urinary bladder for 20 min via an EMDA. The authors concluded that the EMDA group showed greater improvement with better sustained effects. AbobotA/EMDA was shown to be a feasible, reproducible, cost-effective, and pain-free method on an outpatient basis with no need for anesthesia [30]. Schurch et al. used a similar technique to improve analgesia by increasing the analgesic effect of the pretherapeutic intravesical lidocaine solution with EMDA. This procedure reduced the pain level from an average of 4 (on a 10-point rating scale) to an average of 0.5 [31].

Reducing the number of injections should reduce the pain during this procedure and lead to better acceptance of the therapy. However, Chang et al. found different results in a study attempting to reduce postprocedural pain associated with 5 versus 20 intradetrusor injections. Other than shortening the operation time, the authors found no advantages for the patients. The average pain score was not statistically significant between groups [32]. Similarly, Zdroik et al. found no significant difference in the mean pain score when comparing 10 vs. 20 injections (4 (1.5–5) for 10 injections vs. 3 (1–4) for 20 injections) [33]. Miceli et al. found that procedural discomfort related to botulinumtoxinA injection for idiopathic OAB did not differ between groups administered 5 mL/5 injections and 10 mL/10 injections [34]. This differed from the study by DiCarlo-Meacham, which found that patients’ willingness to undergo OnabotA-DI again was influenced by a reduced number of injections (odds ratio = 3.8 (*p* = 0.004)) [35]. Overall, after reviewing the results in the literature, the reduction in patients’ perceptions of pain through a reduced number of injections appears to be relatively small. In order to generally reduce patients’ fear of the consequences of this procedure, a current meta-analysis in this journal confirms that the therapy has a low and only temporary rate of side effects [36]. 

To our knowledge, there are currently no studies on the influence of different needle thicknesses, lengths, and bevels on pain during botulinumtoxinA detrusor injection. Our study shows a statistically significant advantage in favor of using a thinner needle.

In summary, our study shows the following:The overall pain of the OnabotA-DI is mild to moderate; the VAS pain score is, on average, between 2.1 and 2.8.However, there are patients for whom the procedure should be performed under general anesthesia. This is shown by isolated patients who reported pain in the upper third of the VAS.A rising pain sensitization during the procedure appears to occur quickly, so the procedure should be kept short (via a reduction in injection sites).The choice of needle is of (at least moderate) importance regarding the sensation of pain.

The difference in the pain score is not a groundbreaking factor for the clinical decision regarding needle selection, as other user aspects such as length, handling, cannula thickness (which must fit through the cystoscope used), the stiffness of the cannulas, and price aspects also influence the choice of the most suitable injection needle.

However, if medical professionals have two or more injection needles of the same quality and no other advantages, they might consider to use the injection cannula with the thinnest needle according to this study.

## 4. Summary

The pain caused by the OnabotA-DI remains an issue that needs to be mitigated to make the procedure less stressful and increase patient acceptance. There is no clear answer, but there are a variety of nuances that may lead to a less painful procedure. The personal perception of pain among patients should not be underestimated; this is certainly important in all of the optimization efforts carried out. Factors that might make the procedure less unpleasant include the alkalinization of the anesthetic instillation, a reduction in the number of injection sites, concomitant systemic analgesia, and using the smallest possible needle gauge.

## 5. Material and Methods

Female patients with iOAB or spontaneously voiding patients with multiple sclerosis whose standard treatment had failed decided to undergo OnabotA-DI. They were asked to take part in this single-blind study. Each patient’s written consent for the procedure was obtained. These were first or repeat injections. The study protocol was approved by the ethics committee of the Charité Universitätsmedizin Berlin (number EA4/203/22 from 9 January 2023). In accordance with current EAU guidelines, in addition to the general medical history, a sonographic residual urine determination was carried out preoperatively, a urinary tract infection was ruled out using urine sediment and culture, and a vaginal examination and a cystoscopic examination ruled out a tumor, stone, or other pathology of the pelvic floor or vagina [5].

Anticoagulant medication was stopped two days before the procedure according to the OnabotA medication information sheet. An antibiotic, generally trimethoprim 200 mg, was administered perioperatively on the evening of the injection day and the next morning. After the urinary bladder was emptied on the toilet immediately before the procedure, the procedure was carried out in the lithotomy position; after disinfection of the urethral opening, the urinary bladder was emptied with a disposable catheter (thus excluding residual urine), and anesthesia was carried out by instilling lidocaine solution 2%, 50 mL. This solution was left for 20 min.

The injection was carried out using a standardized 21 Char rigid cystoscope from Wolf with an Albaran insert and under optical control via a digital monitor, which could also be viewed by the patient if desired. The choice of needle was randomized based on availability due to supply restraints during and after the COVID-19 pandemic. Three different products were used for injection cannulas or needles, which differed in cannula length and needle length, thickness, or company-specific cut (needle A: length 4 mm, thickness 22 G; needle B: length 5 mm, thickness 27 G; needle C: length 4 mm, thickness 22 G, sharp cut 15°). The preparation used was OnabotA (Botox^®^, AbbVie, Irvine, CA, USA) in doses between 100 and 200 units. Regardless of the number of units, the vials were drawn up to 10 mL sodium chloride solution 0.9% in a 10 mL syringe. After 20 min of exposure to local anesthesia, the urinary bladder was filled to about 2/3 of its maximum capacity, enough to create sufficient wall resistance but not so much as to cause discomfort or pain due to excessive filling. The needles were systematically injected into the detrusor muscle at 17 locations (side walls, posterior wall, bladder roof, bladder base, and co-injection of the trigone, as shown in Figure 1). The reduction in the number of injections from 20, as described in the package leaflet, to 17, which was standardized in this study, was carried out to reduce the patients’ pain burden [35]. This procedure was performed in exactly the same way in neurogenic and non-neurogenic patients.

For each injection, patients rated their pain on a VAS of 1–10 (1 = no pain; 10 = worst pain imaginable). After the procedure, the urinary bladder was emptied, and the patients were observed for half an hour post-intervention for well-being, pain, or hematuria. We did not measure the duration of effect and satisfaction with the injection in these patients as this was not the objective of this study. For statistical analyses, the mean, standard deviation, median, and range were calculated for metrically scaled variables (age, blood pressure, etc.). Absolute and relative frequencies were calculated for nominally scaled variables.

To test the three needle types for differences in the variables examined, one-way analysis of variance (with prior testing for normality), Fisher’s exact test, or chi-squared tests were used depending on the variable level. For pairwise comparisons, the resulting *p*-value was adjusted according to Bonferroni.

## Figures and Tables

**Table 1 toxins-16-00395-t001:** Age, diagnoses, and first/repeat injections of patients.

Characteristic	Overall n = 68	Needle A (22 G, 4 mm) n = 33 ^3^	Needle B (27 G, 5 mm) n = 20 ^3^	Needle C (22 G, 4 mm, Sharp Cut 15°) n = 15 ^3^	*p*-Value ^2^
** Age **					0.6
Mean (SD)	64.0 (15.9)	62.1 (15.3)	64.8 (15.3)	66.9 (18.4)	
Median	66.0	64.0	67.5	76.0	
Range	27.0–86.0	27.0–86.0	36.0–83.0	29.0–85.0	
** Diagnose **					0.3
iOAB ^1^	60 (88.2%)	28 (84.8%)	17 (85.0%)	15 (100.0%)	
nDO in MS ^1^	8 (11.8%)	5 (15.2%)	3 (15.0%)	0 (0.0%)	
** Injection **					0.6
First injection	40 (58.8%)	21 (63.6%)	10 (50.0%)	9 (60.0%)	
Repeat injection	28 (41.2%)	12 (36.4%)	10 (50.0%)	6 (40.0%)	

^1^ Idiopathic overactive bladder; neurogenic detrusor overactivity (nDO) in patients with multiple sclerosis. ^2^ One-way ANOVA; Fisher’s exact test; Pearson’s chi-squared test. ^3^ Different frequencies of needles due to delivery bottlenecks during and after COVID-19 pandemic (see text).

## Data Availability

The raw data supporting the conclusions of this article will be made available by the authors on request.

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
