# Peer review of "Assessing the Use of BotulinumtoxinA for Hyperactive Urinary Tract Dysfunction a Decade after Approval: A Single-Blind Study to Evaluate the Reduction in Pain in OnabotulinumtoxinA Detrusor Injection Using Different Injection Needles"

_toxins, 2024, doi:10.3390/toxins16090395_

Round 1

Reviewer 1 Report

Comments and Suggestions for Authors

This is a nice study and result relevant for clinical practice. My only concern is the tables as they are not understandable without reading the text.

Tables have to be self-explanatory:

-          Plenty of abbreviations not explained in the legend

-          Why using needle 1, 2 and 3? It is better to write needle 27G,….

-          M1-RS1 in the Tables, refer to Figure 1 in the legend

-          All Tables need a legend, also Table 3.

No information on differences between areas of injection in the bladder in the abstract and almost nothing in the result section with exception of the large tables. Nothing on the results of site of injection in the discussion section although there are large Tables + Figure about it in the results?

The authors must improve the Tables and add more legend and need to discuss the impact of site on injections. Furthermore they need to say something about the large bulk of results on site of injection in their abstract and discussion.

Reviewer 2 Report

Comments and Suggestions for Authors

Interesting study!...

Comments

Why was not used a flexible cystoscope with its needles?

What was the local anesthesia (20 min) used?

There are different number of injections between neurogenic and non neurogenic patients as was approved, you used 17 to decrease pain so...

Important also to measure the duration of effect and satisfaction amongst these patients (quite different to those who we use a light sedation, I suppose)

Reviewer 3 Report

Comments and Suggestions for Authors

Reviewer 4 Report

Comments and Suggestions for Authors

I have gone through manuscript entitled “Botulinumtoxin for Hyperactive Urinary Tract Dysfunction - after a Decade from Approval: Needle Matters: A Single-Blinded Study to Evaluate the Pain Reduction of OnabotulinumtoxinA Detrusor Injection Using Different Injection Needles” and found interesting. Some of minor corrections are:

1. author should add the schematic representation of whole study in introduction section so audience can easily understand the hypothesis.

2. Author should include recent reference papers for the study.

3. author mentioned different reference in discussion part that creates confusion, please revisit and put only relevant one.

4. In summary sections, in place of bullet points author can show his output by graphical way for better understanding the study.

5.  some abbreviation are missing, please revisit and add.

Round 2

Reviewer 2 Report

Comments and Suggestions for Authors

No comments

Thank you very much for your revisions 

Comments on the Quality of English Language

No comments

Thank you very much for your revisions